**a** | **Open Peer Review** | Parasitology | Research Article

# RPA-CRISPR/Cas12a-LFA combined with a digital visualization instrument to detect *Toxoplasma gondii* in stray dogs and cats in Zhejiang province, China

Hao Sun,[1] Jiyuan Fan,[1] Hongkun Chu,[1] Yafan Gao,[1] Jiawen Fang,[1] Qinli Wu,[1] Haojie Ding,[1,2,3] Xunhui Zhuo,[1,2,3] QingMing Kong,[1,2,3] HangJun Lv,[1,2,3] Bin Zheng,[1,2,3] Shaohong Lu[1,2,3]

**ABSTRACT** *Toxoplasma gondii*, which causes toxoplasmosis, is prevalent in warm-blooded animals, such as cats, dogs, and humans. *T. gondii* causes economic losses to livestock production and represents a potential risk to public health. Dogs and cats are common hosts in the epidemiology of toxoplasmosis. The current molecular diagnostic tools for *T. gondii* infection require high technical skills, a laboratory environment, and complex instruments. Herein, we developed a recombinase polymerase amplification (RPA)-clustered regularly interspaced short palindromic repeats (CRISPR)/CRISPR-associated protein 12a (Cas12a) assay to detect *T. gondii*. The lowest limit of detection of the assay was 31 copies/µL for the *T. gondii B1* gene. In addition, we established a visual RPA-CRISPR/Cas12a lateral flow band assay (RPA-CRISPR/Cas12a-LFA) combined with a digital visualization instrument, which minimized the problem of false-negative results for weakly positive samples and avoided misinterpretation of the results by the naked eye, making the LFA assay results more accurate. The assay established in this study could identify *T. gondii* within 55 min with high accuracy and sensitivity, without cross-reaction with other tested parasites. The developed assay was validated by establishing a mouse model of toxoplasmosis. Finally, the developed assay was used to investigate the prevalence of *T. gondii* in stray cats and dogs in Zhejiang province, Eastern China. The positive rates of *T. gondii* infection in stray cats and dogs were 8.0% and 4.0%, respectively. In conclusion, the RPA-CRISPR/Cas12a-LFA is rapid, sensitive, and accurate for the early diagnosis of *T. gondii*, showing promise for on-site surveillance.

**IMPORTANCE** *Toxoplasma gondii* is a virulent pathogen that puts millions of infected people at risk of chronic disease reactivation. Hosts of *T. gondii* are distributed worldwide, and cats and dogs are common hosts of *T. gondii*. Therefore, rapid diagnosis of early *T. gondii* infection and investigation of its prevalence in stray dogs and cats are essential. Here, we established a visual recombinase polymerase amplification-clustered regularly interspaced short palindromic repeats (CRISPR)/CRISPR-associated protein 12a-assay combined with a lateral flow band assay and a digital visualization instrument. Detailed analyses found that the assay could be used for the early diagnosis of *T. gondii* without false-negative results. Moreover, we detected the prevalence of *T. gondii* in stray cats and dogs in Zhejiang province, China. Our developed assay provides technical support for the early diagnosis of *T. gondii* and could be applied in prevalence surveys of *T. gondii* in stray dogs and cats.

**KEYWORDS** *Toxoplasma gondii*, diagnosis, RPA-CRISPR/Cas12a, LFA, prevalence

T*oxoplasma gondii* is an obligate intracellular protozoan from the phylum Apicomplexa, which is an opportunistic pathogen with the capacity to infect nearly all

Address correspondence to Shaohong Lu, llsshh2003@163.com, or Bin Zheng, bin_zheng@foxmail.com.

The authors declare no conflict of interest.

warm-blooded animal species, causing toxoplasmosis (1). *T. gondii* is widely distributed around the world, and approximately 30% of the human population is seropositive for *T. gondii* (2). It causes significant economic losses in animal husbandry and has adverse effects on food safety (3, 4). Recent studies showed that the rates of *T. gondii* infection in the general population and pregnant women were 8.20% and 8.60%, respectively, in China (5, 6). In some developing countries, the rate of congenital toxoplasmosis is higher than in developed countries. For example, Brazilian children have a five times higher risk of severe toxoplasmosis than children in Europe (7). Infections with *T. gondii* can lead to a variety of problems, including malaise, fever, and cervical lymphadenopathy or chorioretinitis. Importantly, *T. gondii* can cause a variety of diseases in people of all ages, leading to congenital malformations in pregnant women and multiple complications and death in immunodeficient individuals (8, 9). Both humans and pets (dogs and cats) are susceptible to toxoplasmosis. Cats are not only the definitive host of *T. gondii* but also excrete the environmentally resistant oocysts in their feces (10). Millions of *T. gondii* oocysts can be shed from felids, leading to infection of other susceptible hosts (11). Meta-analyses over the past 50 years showed that the seroprevalence of *T. gondii* in domestic and wild felids was around 35% and 59%, respectively (12). In China, about 20.3% of cats and 11.1% of dogs are seropositive for *T. gondii* (13, 14). With rapid economic and social development, pet ownership is becoming more common, and the number of stray cats and dogs is also increasing indirectly, which enhances the risk of human toxoplasmosis. Unfortunately, there are no protective vaccines or medicines against cat or human toxoplasmosis (15). Therefore, the development of a highly specific and sensitive method for the early diagnosis of *T. gondii* infection is essential to maintain human safety and control the transmission of toxoplasmosis.

Various methods have been used to diagnose toxoplasmosis, including pathogen detection, immunological techniques, and molecular biological methods. Microscopic detection of tachyzoites or tissue cysts is a direct diagnostic test. However, this method is time-consuming, has low sensitivity, and might result in misdiagnosis (16). Serological diagnostic methods based on the detection of specific antibodies for *T. gondii* are the most commonly used diagnostic methods (17). However, the window to detect the risk of *T. gondii* infection was significantly longer, particularly when antibody testing was used, with a *T. gondii* window period of approximately 2 months, and the determination of an antibody subtype is not sufficient to prove *Toxoplasma gondii* infection. Therefore, serological diagnosis might carry a risk of false-negative results in the early stages of *T. gondii* infection (18, 19). Furthermore, differences in the antigens used in the kits comprising clinical *T. gondii* test strip testing can also lead to different detection rates of IgG and IgM in patients (20). Polymerase chain reaction (PCR), quantitative real-time PCR (qPCR), and loop-mediated isothermal amplification (LAMP) assays have played increasingly important roles in the molecular diagnosis of *T. gondii* (21, 22). However, qPCR and PCR require a thermal cycler, which limits their on-site application (23). The primer design of LAMP is complicated, and LAMP can easily cause aerosol pollution, resulting in false positives (24). Recombinase polymerase amplification (RPA) is an isothermal amplification method that works between 37°C and 42°C and can be used to detect nucleic acids rapidly and without the need for complex laboratory equipment. It can detect very low concentrations of pathogen-specific nucleic acids in 20 min (25). RPA has been developed to detect several pathogens, such as *Francisella tularensis*, *Yersinia pestis*, *Bacillus anthracis*, and *variola virus* (26). However, aerosol pollution or nonspecific amplification of the primers and probe can cause false-positive results in RPA detection systems. The clustered regularly interspaced short palindromic repeats (CRISPR)/ CRISPR-associated protein (Cas) detection technology targets and cleaves specific nucleic acid sequences to finally detect the target sequence. The commonly employed Cas nucleases that cleave DNA include Cas9 and Cas12, which exhibit non-specific cleavage activity when they bind their specific targets (27). Compared with Cas9, Cas12 requires a shorter CRISPR RNA (crRNA) and has an additional trans-cleavage activity to cleave

single-stranded DNA with an arbitrary sequence, which has led to Cas12 being more widely used in the field of genetic engineering in recent years. Combining CRISPR/Cas with nucleic acid amplification methods such as RPA, LAMP, and PCR could improve the sensitivity of detection and compensate for the risk of false positives in nucleic acid amplification methods (28). CRISPR-Cas12a/Cas13a systems have been applied to detect different parasites, including *Cryptosporidium parvum* and *Plasmodium* (29–32). For *in situ* applicability and convenience, an RPA-CRISPR/Cas12a system is often combined with a lateral flow band assay (LFA). However, test strips named "elimination method," which have been used for RPA-CRISPR/Cas12a-LFA, have the disadvantage of false negatives for weakly positive samples (33). Incompletely cut single-stranded DNA will form bands of different colors at the T-line of the test strip. Using a digital visualization instrument to read the T-line allows positive samples to be determined using instrument readings, which can avoid the misreading of the results by eye.

Therefore, for the convenient and early diagnosis of *T. gondii*, we aimed to target the *T. gondii B1* gene and establish a sensitive and simple assay based on RPA-CRISPR/Cas12a. To optimize the accuracy of the assay in the field, we combined RPA-CRISPR/Cas12a-LFA with a visualization and digitization instrument to solve the problem of false-negative results from weakly positive samples on the test strips. First, we designed a crRNA probe for the CRISPR/Cas12a detection system in combination with RPA by targeting the *B1* gene of *T. gondii*, which was termed the RPA-CRISPR/Cas12a detection system. This was further improved by incorporating it into an LFA (RPA-CRISPR/Cas12a-LFA). Second, we improved the detection limit of RPA-CRISPR/Cas12a-LFA and optimized the accuracy of data reading by reading and plotting the T-line values of the test strips after visualization and after using the digitization instrument. Third, we applied the developed method to screen the prevalence of *T. gondii* in blood samples of stray dogs and cats in Zhejiang province. Our data showed that the RPA-CRISPR/Cas12a-LFA combined with a digital visualization instrument has the advantages of sensitivity, a short detection window, and portability for the detection of *T. gondii*. Besides, screening of cat and dog samples using the new method might provide guidance for the prevention and control of toxoplasmosis in Zhejiang province. Moreover, the application of the RPA-CRISPR/Cas12a-LFA detection system might play a major role in the early detection of *T. gondii* infection and its prevention.

## RESULTS

### Establishment of the RPA-CRISPR/Cas12a detection technology

The principles of RPA-CRISPR/Cas12a and RPA-CRISPR/Cas12a-LFA are shown in Fig. 1. We used the *T. gondii* whole genome as a template to amplify the *B1* gene by PCR using primers *B1*-F and *B1*-R (Table 1; Fig. 2A) and cloned the gene to obtain plasmid pMD-19T-*B1*. The specificity of the RPA primers was screened using agarose gel electrophoresis analysis (Fig. 2B). The agarose gel electrophoresis results showed that primer sets 3 (RPA-3-F/R), 4 (RPA-4-F/R), 6 (RPA-6-F/R), 7 (RPA-7-F/R), 8 (RPA-8-F/R), 9 (RPA-9-F/R), 10 (RPA-10-F/R), and 12 (RPA-12-F/R) presented single and clear bands between 100 and 200 bp, with no band in the negative control group. Thus, we used the above-selected primers and their corresponding crRNAs (Table 1) for further study. In the RPA-CRISPR/Cas12a system, 1 pg of vector pMD-19T-*B1* was used as the template. As shown in Fig. 3A, the time of onset of the peaks of primer set 8 (F8 + R8) and its corresponding crRNA-2 was earlier than that of the other primer sets. Consequently, primers RPA-8-F/R and crRNA-2 were selected for subsequent experiments. To determine the best temperature for RPA-CRISPR/Cas12a, eight RPA-CRISPR/Cas12a reactions were carried out at 30℃, 32℃, 34℃, 36℃, 37℃, 38℃, 39℃, and 40℃. Analysis of the RPA-CRISPR/Cas12a results (Fig. 3B) showed that although fluorescence signals were produced at all eight temperatures, the earliest peak rise time and strongest fluorescence signals were obtained when the temperature was 40℃. Therefore, 40℃ was chosen as the optimal temperature for the RPA-CRISPR/Cas12a detection experiments.

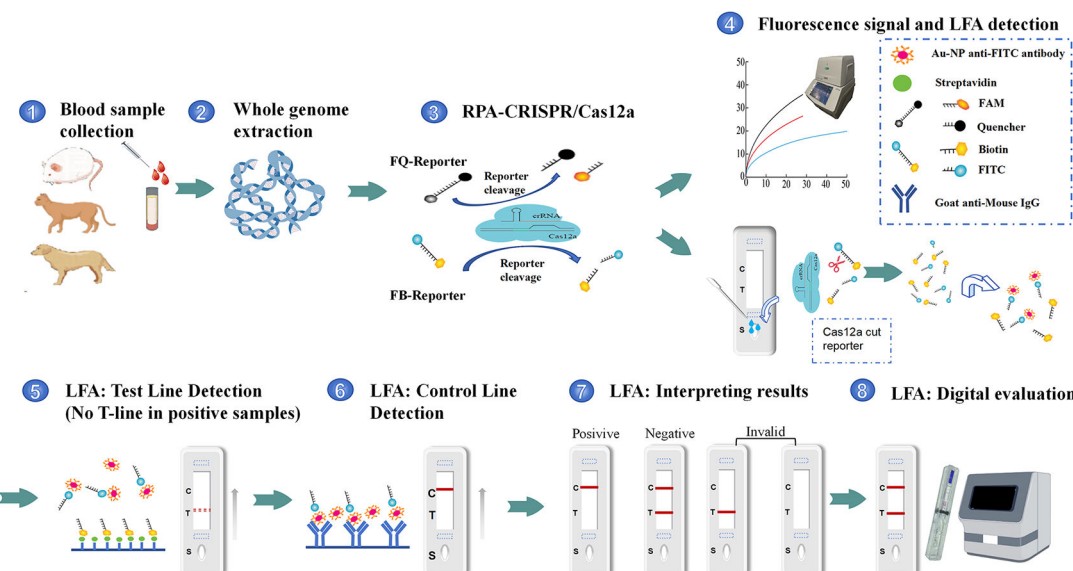

**FIG 1** Schematic diagram of the RPA-CRISPR/Cas12a and RPA-CRISPR/Cas12a-LFA-based detection of *T. gondii*. Genomic DNA extracted from blood was amplified by recombinase polymerase amplification. On positive samples, when the Cas12a combined with the crRNA sequence, the Cas12a endonuclease was activated, trans-cutting the FAM-Quencher (FQ) reporter and the fluorescein isothiocyanate (FITC)-Biotin (FB) reporter in the system, leading to FAM fluorescence becoming separated from the Quencher and was thus detected by the qPCR instrument. In addition, FITC combined with the Au-NP anti-FITC cannot be captured by streptavidin causing no band to appear in the Test line (T-line). Finally, we used a digital visualization instrument to read the T-line values, which improved the sensitivity of the RPA-CRISPR/Cas12a-LFA and avoided false-negative results that could be misinterpreted by the naked eye.

## RPA-CRISPR/Cas12a has robust sensitivity and specificity to detect *T. gondii*

Sensitivity analysis of RPA-CRISPR/Cas12a was performed using different concentrations of vector pMD-19T-*B1*. The statistical analysis revealed that the plasmid at concentrations ranging from 1 to $10^{-5}$ ng had significantly higher fluorescence intensity than that of the negative control group and concentrations of $10^{-6}$ to $10^{-7}$ ng also had higher fluorescence intensity than that of the negative control group (Fig. 3C); however, there was no difference in fluorescence intensity between the plasmid at $10^{-8}$ ng and the negative control group. The results showed that the lower limit of detection was 31 copies/µL, and the plasmid copy number was calculated using the equation: DNA (copy number) = [(6.023 × 1,023) × (copy number/mol) × DNA amount (g)]/[DNA length (bp) × 660 × (g/mol/bp)], where the base number of recombinant plasmid pMD-19T-*B1* was 4,907 bp (34). The sensitivity of RPA-CRISPR/Cas12a to detect pMD-19T-*B1* was 10 times greater than the sensitivity of qPCR (Fig. 3D) and 100 times greater than the sensitivity of PCR (Fig. 3E).

The specificity of RPA-CRISPR/Cas12a detection against different parasite DNA samples was also analyzed, such as those from *Giardia lamblia*, *Cryptosporidium parvum*, *Enterocytozoon bieneusi*, *Blastocystis hominis*, and *Neospora caninum*. As shown in Fig. 3F, only *T. gondii* showed a high fluorescence signal; the samples from the other parasites and the negative control (N) showed no signals. These results demonstrated that RPA-CRISPR/Cas12a detection had sufficient specificity.

## Evaluation of the sensitivity and specificity of RPA-CRISPR/Cas12a-LFA

The principle of RPA-CRISPR/Cas12a-LFA is shown in Fig. 4. *G. lamblia*, *C. parvum*, *E. bieneusi*, *B. hominis*, *N. caninum*, and *T. gondii* samples were tested using RPA-CRISPR/Cas12a-LFA. The *G. lamblia*, *C. parvum*, *E. bieneusi*, *B. hominis*, and *N. caninum* samples showed a clear band at the T-line like that of the negative control but no visible band at the T-line as in the *T. gondii* samples. These results confirmed the specificity of the RPA-CRISPR/Cas12a-LFA assay (Fig. 5A). We further evaluated the sensitivity of

**TABLE 1** List of primers and crRNAs used in the present study[a]

| | Name | Sequence (5'–3') |
|---|---|---|
| 1 | **B1-F** | **TGAGGTCATATCGTCCCATG** |
| | **B1-R** | **CAAGAATGTTGCATTCTTCA** |
| 2 | RPA-1-F | GTTGTCATGCCATCGACGTAGACCCAGAAAT |
| | RPA-1-R | GTGGTGGGCTCGTTGATGAAGTAATCCATTT |
| 3 | RPA-2-F | GCATTGCCCGTCCAAACTGCAACAACTGCTC |
| | RPA-2-R | AGAGTTCGTCGGTGTTTGCTGGGTTGGCTGA |
| 4 | RPA-3-F | ATTGCCCGTCCAAACTGCAACAACTGCTCTA |
| | RPA-3-R | TTTGCTGGGTTGGCTGAAAGATAGGAGGGAG |
| 5 | RPA-4-F | GCATTGCCCGTCCAAACTGCAACAACTGCTC |
| | RPA-4-R | CGTCGGTGTTTGCTGGGTTGGCTGAAAGAT |
| 6 | RPA-5-F | ATTGCCCGTCCAAACTGCAACAACTGCTCTA |
| | RPA-5-R | CGTCGGTGTTTGCTGGGTTGGCTGAAAGATA |
| 7 | RPA-6-F | GCAACAACTGCTCTAGCGTGTTCGTCTCCAT |
| | RPA-6-R | AGAGTTCGTCGGTGTTTGCTGGGTTGGCTGA |
| 8 | RPA-7-F | ACTGCTCTAGCGTGTTCGTCTCCATTCCGTA |
| | RPA-7-R | AGAGTTCGTCGGTGTTTGCTGGGTTGGCTGA |
| 9 | **RPA-8-F** | **GTGAAACAATAGAGAGTACTGGAACGTCGCCGC** |
| | **RPA-8-R** | **GCATGGTTTGCACTTTTGTGGTTTAGCCTCTCG** |
| 10 | RPA-9-F | GAGAGGCTAAACCACAAAGTGCAAACCATGCG |
| | RPA-9-R | GCGGCTACTTTAGAAACGCTTTAGCACTGGGAA |
| 11 | RPA-10-F | AGTTGTCATGCCATCGACGTAGACCCAGAAATG |
| | RPA-10-R | CGAGGCAACCATCACCAACTGCTTTTGTTAAGC |
| 12 | RPA-11-F | GGTCGAGAGGCTAAACCACAAAGTGCAAACCA |
| | RPA-11-R | TGTGCGGCTACTTTAGAAACGCTTTAGCACTGG |
| 13 | RPA-12-F | AAGTTCCGGTCGAGAGGCTAAACCACAAAGTG |
| | RPA-12-R | CCTTAGCATTCCGCATTGTGCGGCTACTTTAGA |
| 14 | crRNA-1 | UAAUUUCUACUAAGUGUAGAUUUCUUUUAGCCUCAAUAGCAG |
| 15 | **crRNA-2** | **UAAUUUCUACUAAGUGUAGAUUGGUUUAGCCUCUCGACCGGA** |
| 16 | crRNA-3 | UAAUUUCUACUAAGUGUAGAUAGAAGGAACUCGAGGCAACCA |

[a]The bold characters indicate the suitable sequences that were screened and used for further study.

RPA-CRISPR/Cas12a-LFA using a serial 10-fold dilution of plasmid pMD-19T-B1. The results showed that the lower limit of detection was 100 fg (310,000 copies/µL) (Fig. 5B).

## Minimizing the disadvantage of false negatives of RPA-CRISPR/Cas12a-LFA using a digital visualization instrument

We discovered that pMD-19T-B1 at $10^{-3}$ to $10^{-4}$ ng developed a clear band in the C-line and no definite band in the T-line, whereas $10^{-5}$ to $10^{-7}$ ng showed a low-strength band compared with $10^{-8}$ ng and the negative control in the T-line, which suggested that the RPA-CRISPR/Cas12a-LFA has the disadvantage of false negatives in weakly positive samples (Fig. 5B) . The low-strength band at the T-line in the RPA-CRISPR/Cas12a-LFA results for weakly positive samples could easily be misread by the eyes and thus affect the accuracy of the resultant readout. The RPA-CRISPR/Cas12a method for the detection of FAM fluorescence using a qPCR instrument showed that the maximum detection limit of the method established in this experiment could reach $10^{-7}$ ng of pMD-19T-B1. Therefore, we believe that the low-strength band for $10^{-7}$, $10^{-6}$, and $10^{-5}$ ng of pMD-19T-B1 in the RPA-CRISPR/Cas12a-LFA detection technology was induced by Cas12a cutting the reporter (FITC-Biotin) of the buffer system incompletely in this weakly positive sample.

To address this issue, we measured the T-line values of the test strips for plasmids at different copy numbers using a digital visualization instrument (Fig. 1 and 5B), and the data were plotted using logarithmic curve fitting to estimate the relationship between B1 gene copy number and the value of the color of the T-line of the RPA-CRISPR/Cas12a-LFA.

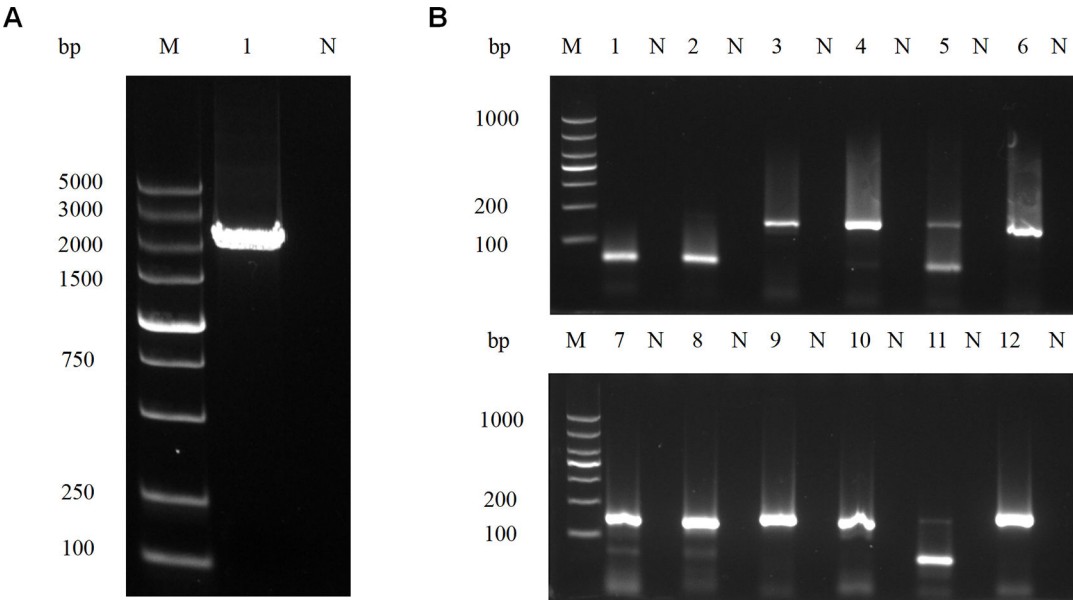

**FIG 2** RPA primers specific to the *B1* gene were screened by PCR. (A) PCR amplification of the *T. gondii B1* gene. M, DNA marker; 1, *T. gondii B1* gene; and N, negative control. (B) Electrophoresis results of different RPA primer pairs after PCR using pMD-19T-*B1* as the template; 1, 2, 3, 4, 5, 6, 7, 8, 9, 10, 11, and 12 represent RPA primer pairs. M, DNA marker and N, negative control. Primer pairs 3, 4, 6, 7, 8, 9, and 10 presented single and clear bands between 100 and 200 bp.

The results showed that the *B1* gene copy number was inversely related to the T-line value (Fig. 5C). Next, we took the T-line value of pMD-19T-*B1* of $10^{-7}$ ng (64) as the critical T-line value for a positive sample test strip. This curve allowed us to estimate the number of copies of the *B1* gene in a sample by measuring the T-line value of the test strip after the sample reaction and to determine whether a sample was negative or positive based on whether the measured T-line value was less than 64 (the T-line value for $10^{-7}$ ng of pMD-19T-*B1*).

Then, simulated *T. gondii* DNA samples were analyzed using the digital visualization instrument. As shown in Fig. 6A, a low-strength band was observed at the T-line in the *T. gondii* DNA sample at $1.6 \times 10^{-2}$ ng (g2). To better observe the color change of the T-line in the DNA samples, we performed serial twofold gradient dilutions of $3.2 \times 10^{-2}$ ng of DNA. As shown in Fig. 6B, $8 \times 10^{-3}$, $1.6 \times 10^{-2}$, and $3.2 \times 10^{-2}$ ng/μL of DNA all had a low-strength band at the T-line, which made it difficult to judge visually whether the sample was negative or positive. The above doubtful test strips were put into the digital visualization instrument for T-line reading, and the instrument displayed a T-line value of 47 for the $8 \times 10^{-3}$ ng/μL of DNA; therefore, it was decided that this sample was positive because the sample T-line was less than the critical T-line value of the test strip for positive samples.

## RPA-CRISPR/Cas12a-LFA can be used to detect early *T. gondii* infection

Samples from mice infected with *T. gondii* were tested using the RPA-CRISPR/Cas12a technique and compared with the results of qPCR detection. The results showed that *T. gondii* DNA was successfully detected in all DNA samples collected at 1, 3, and 5 days, whereas the negative control group injected with saline produced negative results (Fig. 6C). The results of RPA-CRISPR/Cas12a detection were similar to those of qPCR. Second, the DNA samples were tested using the LFA method (Fig. 6D). The experimental results demonstrated that RPA-CRISPR/Cas12a-LFA could detect early *T. gondii* infection, and it could avoid the window period for serological testing.

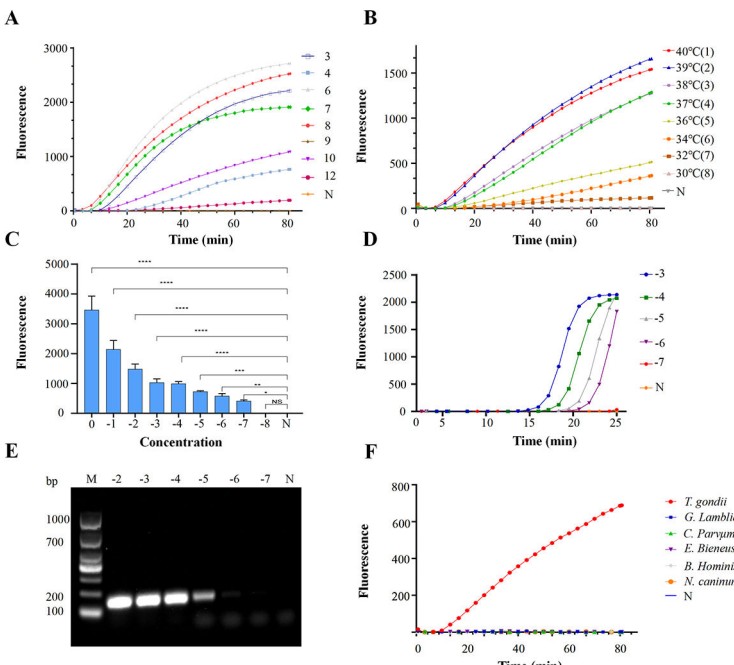

**FIG 3** Establishment of the RPA-CRISPR/Cas12a detection system and sensitivity and specificity analysis. (A) RPA-CRISPR/Cas12a reaction using different primers and crRNAs: 3, RPA-3-F/R-crRNA-1; 4, RPA-4-F/R-crRNA-1; 6, RPA-6-F/R-crRNA-1; 7, RPA-7-F/R-crRNA-1; 8, RPA-8-F/R-crRNA-2; 9, RPA-9-F/R-crRNA-3; 10, RPA-10-F/R-crRNA-2; 12, RPA-12-F/R-crRNA-3; and N, negative control. (B) Optimization of the RPA-CRISPR/Cas12a reaction temperature. 1, 40°C; 2, 39°C; 3, 38°C; 4, 37°C; 5, 36°C; 6, 34°C; 7, 32°C; 8, 30°C; N, negative control. (C) Sensitivity analysis of the RPA-CRISPR/Cas12a assay technique using serial 10-fold dilutions of pMD-19T-*B1* as the template, 0 to (−8) represent 1 ng, 100 pg, 10 pg, 1 pg, 100 fg, 10 fg, 1 fg, 100 ag, and 10 ag; N, negative control. (D) Sensitivity analysis of the qPCR technique using serial 10-fold dilutions of pMD-19T-*B1* as the template, (−3) to (−7) represent 1 pg, 100 fg, 10 fg, 1 fg, and 100 ag; N, negative control. (E) Sensitivity analysis of the PCR technique using serial 10-fold dilutions of pMD-19T-*B1* as a template, (−2) to (−7) represent 10 pg, 1 pg, 100 fg, 10 fg, 1 fg, and 100 ag; N, negative control. ****$P$ < 0.0001; ***$P$ < 0.001; **$P$ < 0.01; *$P$ < 0.1; and NS, not significant. (F) Evaluation of the specificity of the RPA-CRISPR/Cas12a assay.

## Application of the RPA-CRISPR/Cas12a-LFA system to detect the prevalence of *T. gondii* in stray cats and dogs

We examined 248 samples from Deqing, Wenzhou, Jiaxing, Zhoushan, Lishui, and Yiwu cities across Zhejiang province. The RPA-CRISPR/Cas12a method detected *T. gondii* DNA in 7.3% (18/248) and the RPA-CRISPR/Cas12a-LFA method detected *T. gondii* DNA in 5.6% (14/248) of the DNA samples of stray cats and dogs (Fig. 7), and the 14 positive samples in RPA-CRISPR/Cas12a-LFA corresponded with the results of RPA-CRISPR/Cas12a. The positive rates of *T. gondii* in stray cats and dogs in various locations are shown in Tables 2 and 3.

## DISCUSSION

To prevent, screen for prevalence, and control the transmission of *T. gondii* in stray animals, it is necessary to develop a cost-effective and rapid early diagnostic test. Serological testing is a common tool to detect *T. gondii* infection; however, it has the drawback of a longer window period. Moreover, traditional amplification methods, such as PCR and qPCR, are time-consuming, taking up to 2 h or even longer, requiring thermal cycling, and relying on well-established laboratories with sophisticated facilities or well-trained operators (35). Although LAMP is a thermostatic amplification method, its primer design is complex and is prone to aerosol contamination. By

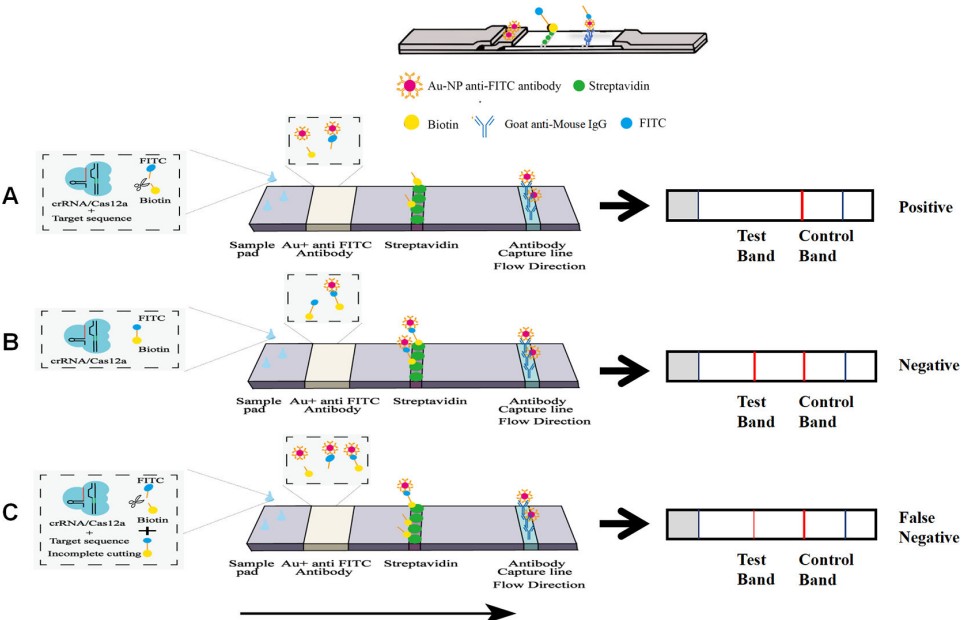

**FIG 4** The "elimination method" strip for CRISPR-based detection. Gold-labeled fluorescein isothiocyanate (FITC)-biotinylated reporter molecules flow to the test capture band and redundant gold nanoparticles flow to the control capture band. (A) In positive samples, activated Cas12a cleaves the FITC-Biotin reporter, causing the separation of FITC from biotin. FITC bound to the Au-NP anti-FITC antibody does not bind to streptavidin at the T-line, which causes no band to appear at the T-line. (B) In negative samples, the FITC-Biotin reporter is not cleaved and binds to the Au-NP anti-FITC antibody, which leads to the T-line showing a band. (C) In weakly positive samples, the presence of incompletely cleaved FITC-Biotin reporter competes for binding to streptavidin at the T-line, forming a lighter-colored T-line, thus weakly positive samples are susceptible to false-negative misinterpretation.

contrast, RPA-CRISPR/Cas12a technology not only has the advantage of early diagnosis by avoiding the window period but also has a simple primer design, high specificity, convenient reaction conditions, and can be performed in a constant temperature water

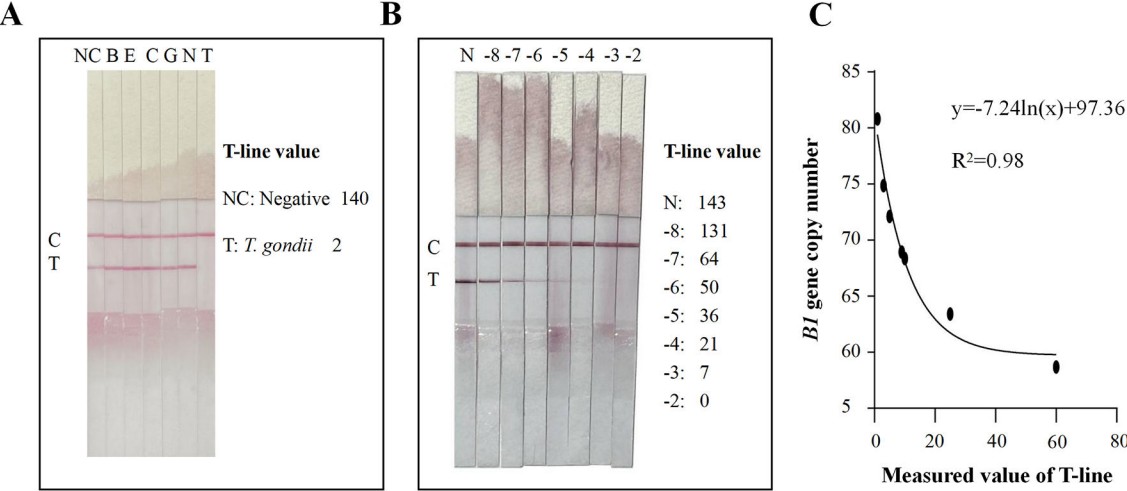

**FIG 5** Establishing the RPA-CRISPR/Cas12a-LFA and conducting digital analysis. (A) Evaluation of the specificity of the RPA-CRISPR/Cas12a-LFA. B, *B. hominis*; E, *E. bieneusi*; C, *C. parvum*; G, *G. lamblia*; N, *N. caninum*; and T, *T. gondii*. (B) Evaluation of the sensitivity of the RPA-CRISPR/Cas12a-LFA using a serial 10 dilution. (−8) to (−2) represent 10 ag, 100 ag, 1 fg, 10 fg, 100 fg, 1 pg, and 10 pg of plasmid pMD-19T-*B1*. (C) Measuring the T-line values of different copy numbers of the plasmid using a digital visualization instrument and plotting the fitted curves.

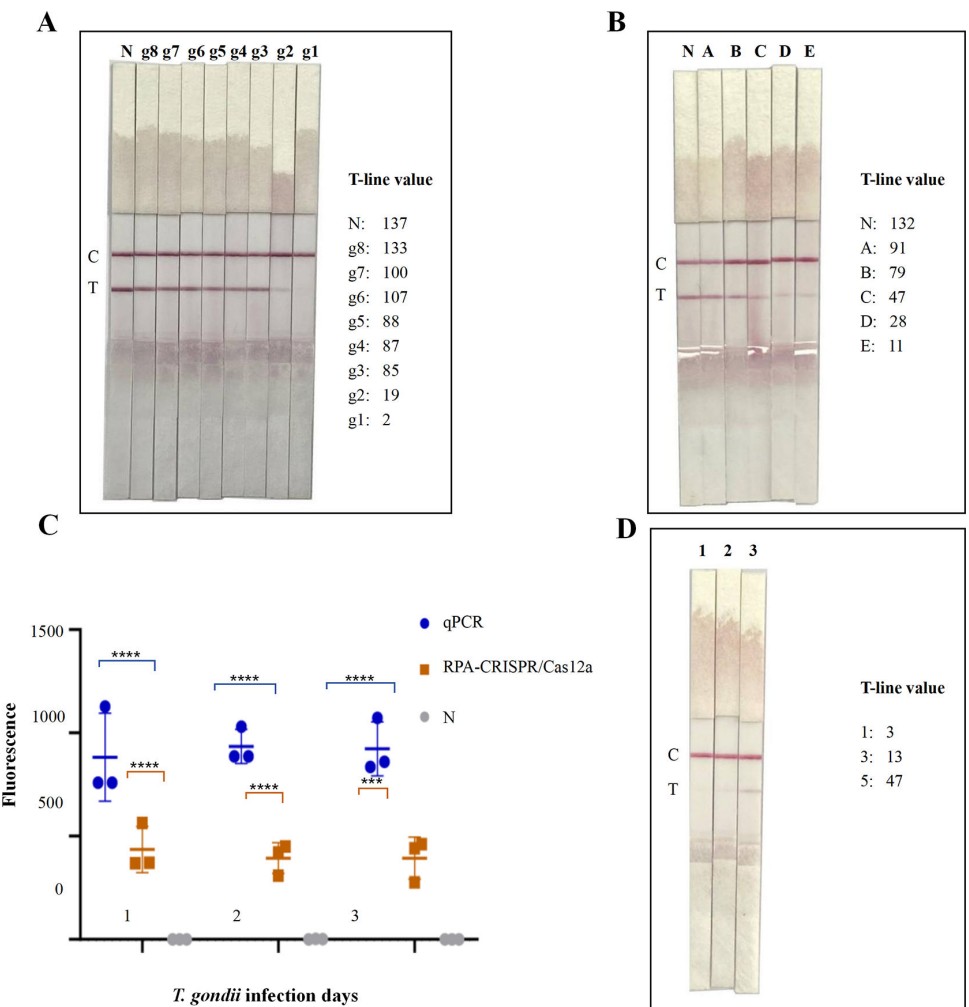

**FIG 6** Analysis results of genomic DNA samples. (A) Evaluation of the sensitivity of RPA-CRISPR/Cas12a -LFA using a serial 10-fold dilution of *T. gondii* genomic DNA. The whole genome is about 1.6 ng/µL. g8–g1 represent $1.6 \times 10^{-8}$ to $1.6 \times 10^{-1}$ ng/µL of *T. gondii* genomic DNA. (B) A–E represent $2 \times 10^{-3}$, $4 \times 10^{-3}$, $8 \times 10^{-3}$, $1.6 \times 10^{-2}$, and $3.2 \times 10^{-2}$ ng/µL of *T. gondii* genomic DNA; N, negative control. (C) Maximum fluorescence signal value from mouse genomic DNA on days 1, 3, and 5 detected using RPA-CRISPR/Cas12a and qPCR; N, negative control. (D) Genomic DNA of infected mice was detected using RPA-CRISPR/Cas12a-LFA. ****$P < 0.0001$ and ***$P < 0.001$.

bath. RPA-CRISPR/Cas12a in combination with LFA test strips (RPA-CRISPR/Cas12a-LFA) is more portable for non-laboratory conditions.

The selection of an appropriate nucleic acid amplification technique combined with the CRISPR-Cas12a system plays a crucial role in clinical diagnosis (36). The RPA assay is accurate, sensitive, and easy to develop within a short period of time. In RPA combined with CRISPR-Cas12a technology, the RPA system can select the target gene and carry out the initial signal amplification (37). The low reaction temperature of RPA is conducive to reducing the aerosol pollution caused by high-temperature reactions (38). Cas12a can also cut non-specific double chains while cutting single chains to reduce reaction contamination (39). In this study, we developed a new assay termed RPA-CRISPR/Cas12a, which combines RPA with CRISPR/Cas12a to detect *T. gondii*. Compared with conventional methods, RPA-CRISPR/Cas12a can detect *T. gondii* in 40 min at 40°C isothermally, without requiring cumbersome thermal cycling steps and expensive equipment. The RPA assay uses a pair of highly specific primers, and the crRNA also directs Cas12a of the CRISPR-Cas12a system to recognize the target DNA, which requires strict selection of the crRNA target to avoid "off-target" effects in genome editing (40). The results

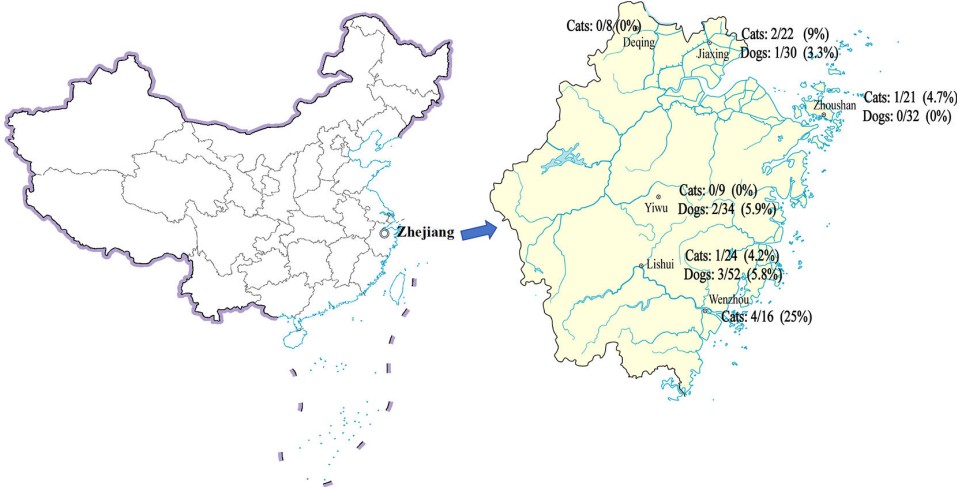

**FIG 7** Rates of *T. gondii* RPA-CRISPR/Cas12a-LFA positivity in the blood of stray cats and dogs in six cities. Map reproduced from China's Ministry of Natural Resource website.

presented here show that the designed crRNA and primers could accurately detect *T. gondii* DNA in the RPA-CRISPR/Cas12a detection system without false-positive signals, confirming the specificity of the detection system for *T. gondii*. Finally, the RPA-CRISPR/Cas12a fluorescent assay has a higher detection sensitivity than conventional assays. The assay had a limit of detection of 31 copies/µL of pMD-19T-*B1*, which was significantly more sensitive than the PCR and qPCR assays.

For the rapid detection of *T. gondii* in field settings, we combined the RPA-CRISPR/Cas12a assay with an LFA to form the RPA-CRISPR/Cas12a-LFA assay system. In previous studies, CRISPR-LFA was reported to be subject to false-negative misreadings in weakly positive samples and was slightly less sensitive when read by the naked eye compared with being read using advanced fluorescent detection devices. To solve this problem, we performed RPA-CRISPR/Cas12a-LFA detection using 10-fold serial dilutions of pMD-19T-*B1* as the templates and observed the color depth of the T-line on the test strip for the different concentrations. A comparison of the results with those of the FAM signal detected using the qPCR instrument suggested that the lighter band on the T-line for $10^{-7}$ ng of pMD-19T-*B1* was a false-negative result caused by incomplete Cas12a cleavage of the reporter (FITC-Biotin) in the weakly positive sample. The T-line values of the test strips were read using a digital visualization instrument, and the curves of *B1* gene copy number (10-fold serial dilutions of pMD-19T-*B1*) and T-line readings were plotted using curve fitting, which allowed us to determine the threshold value of positive samples, thereby improving the sensitivity of RPA-CRISPR/Cas12a-LFA detection. Compared with image signal analysis of T-line signal strength, the instrument used in this study is more convenient and accurate when reading the values (33). In addition, by reading the T-line value using the instrument, we avoided the misinterpretation of

**TABLE 2** Detection of *T. gondii* in blood samples from stray cats and dogs detected using RPA-CRISPR/Cas12a-LFA

| City | Cats | | Dogs | |
|---|---|---|---|---|
| | **Positive samples** | **Total** | **Positive samples** | **Total** |
| Deqing | 0 | 8 | 0 | 0 |
| Jiaxing | 2 (9.0%) | 22 | 1 (3.3%) | 30 |
| Zhoushan | 1 (4.7%) | 21 | 0 | 32 |
| Yiwu | 0 | 9 | 2 (5.9%) | 34 |
| Lishui | 1 (4.2%) | 24 | 3 (5.8%) | 52 |
| Wenzhou | 4 (25%) | 16 | 0 | 0 |
| Total | 8 (8.0%) | 100 | 6 (4.0%) | 148 |

**TABLE 3** RPA-CRISPR/Cas12a and RPA-CRISPR/Cas12a-LFA results for *T. gondii* detection in DNA extracted from blood samples of stray cats and dogs in Zhejiang province, China

| Detection method | Blood samples (*n* = 248) | |
|---|---|---|
| | Positive | Negative |
| RPA-CRISPR/Cas12a | 18 (7.3%) | 230 (92.7%) |
| RPA-CRISPR/Cas12a-LFA | 14 (5.6%) | 234 (94.4%) |

weakly positive samples as judged by the naked eye. Furthermore, in a mouse model of *T. gondii* infection, the experimental group tested positive on the first day, with no positive signals in the negative control (uninfected) group. The developed method can avoid the window period of serological diagnostic testing and thus has the potential to be a powerful tool for the early diagnosis of toxoplasmosis.

Measures to control infectious diseases mainly focus on controlling the source of infection, cutting off the means of transmission, and protecting vulnerable populations. Prenatal screening is an important route in cutting off transmission; however, it is not mandatory in antenatal care (41). Controlling the transmission of *T. gondii* in dogs and cats is also an effective tool to prevent toxoplasmosis. As the economy grows and the standard of living improves, people are acquiring domestic pets or adopting strays as pets (42). Survey data on toxoplasmosis in dogs and cats can inform human prevention and control of toxoplasmosis. Herein, 248 samples from stray cats and dogs were tested using the RPA-CRISPR/Cas12a and RPA-CRISPR/Cas12a-LFA detection systems. Testing using the RPA-CRISPR/Cas12a-LFA detection system revealed 14 positive samples, which was slightly less than the 18 positive samples detected using RPA-CRISPR/Cas12a. The likely reason for this is that the sensitivity of the LFA was slightly lower than that of the qPCR instrument, and the threshold to determine positivity in weakly positive samples is not as precise as it could be, despite the use of the digital visualization instrument. In our study, *T. gondii* infection rates in cats were similar to IgG seroprevalence of *T. gondii* infection in Chinese blood donors in China, which suggested that humans should be advised to take control measures to minimize human-dog/cat contact, thereby reducing exposure to *T. gondii* (43). The prevalence of *T. gondii* infection in the samples from dogs and cats in the eastern part of China used in this study was lower than that reported for the western part of the country, which might be related to the level of economic development and hygiene conditions (44). The DNA positivity rate for dogs and cats was 5.71% in Shanghai in 2012 (45). Our experiment produced similar results and used more experimental DNA samples, which suggested that RPA-CRISPR/Cas12a-LFA is suitable for use in prevalence studies and for diagnosis of the acute stage of feline toxoplasmosis.

In conclusion, we developed a new molecular assay for *T. gondii* that combines CRISPR/Cas12a with RPA pre-amplification and readout using a fluorescence analyzer or LFA. In addition, we used cheaper RPA amplification kits, which greatly reduced the research costs. For on-site test strip testing, we solved the problem of false negatives related to "elimination method" strips by using a digital visualization instrument and plotting a fitted curve, which would improve the accuracy of field sample testing and avoid misinterpretation by the naked eye. The RPA-CRISPR/Cas12a-LFA holds promise as a fast, reliable, and sensitive assay that can be used for detection at a more readily available temperature (e.g., 37°C) or by hand in under field conditions, for the early diagnosis of *T. gondii* without requiring expensive instruments. In addition to the above advantages, RPA-CRISPR/Cas12a-LFA has a promising future in the detection and DNA extraction of *T. gondii*. First, new genetic targets for *T. gondii* could be explored and a multiplexed platform could be established using the CRISPR-Cas system to identify several target molecules in a single reaction, aiming to improve the sensitivity and stability of *T. gondii* detection. Second, a trans-activating crRNA can be engineered as a CRISPR target to type *T. gondii* to shorten the time taken by current PCR techniques. Third, subsequent research should focus on the rapid release of *T. gondii* DNA, simultaneous amplification, and CRISPR cutting of target genes for one-step detection, with the aim of shortening the detection time without compromising detection sensitivity.

Fourth, the probability of stray cats and dogs being infected with *T. gondii* is higher than that of domestic cats and dogs; therefore, it is crucial to actively promote the research and development of pet food or snacks containing *T. gondii* medication to control the transmission of *T. gondii* in stray animals. Finally, in addition to its utility in detecting *T. gondii*, RPA-CRISPR/Cas12a(-LFA) also has promising applications in the study of other parasites.

## MATERIALS AND METHODS

### Experimental mice and samples

All primers used for PCR and qPCR were provided by the Beijing Tsingke Biotech Co., Ltd. (Beijing, China). The crRNAs were synthesized by Sangon Biotech Co., Ltd. (Shanghai, China). The CRISPR/Cas12a DNA Detection Kits and commercial CRISPR Lateral Flow Strip were purchased from EZassay Co., Ltd. (Shenzhen, China). In this study, *T. gondii* tachyzoites (RH strain) were cultured in Vero cells in our laboratory. *G. lamblia*, *C. parvum*, *E. bieneusi*, and *B. hominis* were provided by Zhejiang University, and *N. caninum* was provided by Dr. Jianhua Li from Jilin University to be used for specificity analysis. Six-to-eight-week-old female BALB/c mice were obtained from the Zhejiang Experimental Animal Centre and were housed under standard specific pathogen-free conditions. A total of 248 blood samples from stray dogs and cats were collected from Deqing, Wenzhou, Yiwu, Lishui, and Zhoushan, with the assistance of experts from the animal protection base of the Zhejiang Small Animal Protection Association. All blood samples were stored at 4°C, and nucleic acids were extracted using a TIANamp Virus DNA/RNA Kit (Tiangen, Beijing, China) within 3 days.

### Construction of the positive recombinant plasmid pMD-19T-*B1*

A DNA Quick Extraction Kit (Beyotime, Shanghai, China) was used to obtain the whole genome of *T. gondii*. The 50 µL PCR reaction system included 1 µL of the whole genome of *T. gondii*, 25 µL of 2× PCR buffer for KOD FX, 10 µL of 2 mM dNTPs, 1 µL of primer *B1*-F (10 µM), 1 µL of primer *B1*-R (10 µM), 1 µL of KOD FX, and 11 µL of ddH$_2$O. The amplification procedure was as follows: initial denaturation at 95°C for 5 min, followed by 35 cycles of denaturation (10 s at 98°C), annealing (30 s at 48°C), and extension (150 s at 68°C), with a final extension at 68°C for 10 min. Before ligating the PCR product into the cloning vector, we used a DNA A-Tailing Kit (Takara, Dalian, China) to add A bases to the 3′ end of the PCR product. Then, the A-Tailed DNA fragments were cloned into the pMD19-T vector (Takara). After transformation and selective plating, positive clones were selected and sequenced. The recombinant plasmid was named pMD-19T-*B1*.

### Establishment and optimization of the RPA-CRISPR/Cas12a detection technology

Specific RPA primers for the *B1* gene were designed using Primer Premier 5 software (Premier Biosoft, San Francisco, CA, USA) based on the *B1* gene sequence in GenBank (AF179871.1). PCR was used to select the specific RPA primers for the *B1* gene from among the primer candidates. The PCR reaction system comprised 2 µL of the whole genome of *T. gondii*, 1 µL of primer RPA-F (10 mM), 1 µL of primer RPA-R (10 mM), 12.5 µL of 2× Taq PCR Mix, and 8.5 µL of ddH$_2$O. The PCR products were analyzed using 1% agarose gel. According to the results, we selected the primers corresponding to the single and clear bands. In addition, we designed the crRNA probes based on the screened RPA primers (Table 1). The crRNAs corresponding to RPA-3/4/6/7-F/R, RPA-8/10-F/R, and RPA-9/12-F/R were crRNA-1, crRNA-2, and crRNA-3, respectively. To determine the optimal temperature, RPA-CRISPR/Cas12a primers, and crRNA, we screened the reactions according to the following reaction using RPA-CRISPR/Cas12a: 10 µL of detection buffer (2×) containing FAM fluorescent probe, 5 µL of core mix (4×), 0.5 µL of RPA-F (20 µM), 0.5 µL of RPA-R (20 µM), 0.4 µL of crRNA (2.5 µM), 1 µL of

Cas12a Protein (1 µM), 0.6 µL of pMD-19T-*B1* (1.67 × 10$^{-3}$ ng/µL), and 2 µL of starter (10×). The fluorescence detection process was performed under the following conditions: 120 cycles at 37°C for 30 s. The above reaction system was then used for fluorescence qPCR using the FAM system for the real-time detection of 1 pg of pMD-19T-*B1*. In the RPA-CRISPR/Cas12a experiment, we used water as the negative control to exclude aerosol contamination and avoid false positives. For all these procedures, pMD-19T-*B1*, Cas12a protein, and crRNA were diluted using diethyl pyrocarbonate (DEPC)-treated water (Vazyme, Suzhou, China) according to the experimental requirements. Finally, we determined the optimal primers and temperature for RPA-CRISPR/Cas12a by analyzing the quantification cycle (Cq) values and the height of the fluorescence signal.

## Sensitivity and specificity analysis of RPA-CRISPR/Cas12a detection technology

To test the sensitivity of RPA-CRISPR/Cas12a, we 10-fold serially diluted pMD-19T-*B1*. These dilutions ranged from 1 ng to 10 ag, with nuclease-free water included as a negative control. The experimental results were compared with those derived using qPCR and PCR. To validate the accuracy of the results, each concentration of pMD-19T-*B1* in RPA-CRISPR/Cas12a was tested three times. The optimal Cq values and the height of fluorescent signals were used to analyze the minimum detection limit of RPA-CRISPR/Cas12a. The RPA-CRISPR/Cas12a protocols and procedures were also executed to verify the specificity of the system. Briefly, the targets were detected using the genomes of five different parasite species: *G. lamblia*, *C. parvum*, *E. bieneusi*, *B. hominis*, and *N. caninum*.

## RPA-CRISPR/Cas12a combined with a lateral flow strip assay

To detect the samples quickly and portably, we used the RPA-CRISPR/Cas12a system in conjunction with colloidal gold test strips. The principle of the RPA-CRISPR/Cas12a-LFA detection of *T. gondii* DNA comprised a sample pad, a conjugation pad [with colloidal Au-nanoparticle (NP)-anti-fluorescein isothiocyanate (FITC) antibody], a T-line (with streptavidin), a C-line (with goat anti-mouse IgG), and an absorption pod. Based on the siphon effect of the RPA-CRISPR/Cas12a-LFA, liquid dropped onto the pad is pushed through the conjugation pad, T-line, and C-line in that order. In positive samples (Fig. 4A), Cas12a engages in trans-cleavage activity when it detects the target nucleotide amplified by RPA and cleaves the reporter DNA (FITC-biotin). When the reporter DNA is cleaved, FITC bound to colloidal gold can separate from biotin, thus the Au-NP-anti-FITC-FITC complex, Au-NP-anti-FITC, and biotin mixture are present in the reaction system. At the T-line, streptavidin cannot capture the Au-NP-anti-FITC-FITC complex and Au-NP-anti-FITC; therefore, no band will appear at the T-line, whereas the Au-NP-anti-FITC-FITC complex and Au-NP-anti-FITC can be captured by the goat anti-mouse IgG and form a band at the C-line. By contrast, in negative samples (Fig. 4B), because there is no cleavage of the reporter DNA, the reaction system contains Au-NP-anti-FITC-FITC-biotin complex and Au-NP-anti-FITC. As the reaction system flows to the T-line, the Au-NP-anti-FITC-FITC-biotin complex is captured by streptavidin to form a band, and at the C-line, Au-NP-anti-FITC is captured by goat anti-mouse IgG to form a band. Unfortunately, in weakly positive samples (Fig. 4C), because Cas12a cannot cleave the reporter completely, the reaction system contains a mixture of Au-NP-anti-FITC-FITC complex, Au-NP-anti-FITC, and a small amount of uncleaved Au-NP-anti-FITC-FITC-Biotin complex. When the reaction system flows through the T-line, a small amount of the Au-NP-anti-FITC-FITC-Biotin complex will be captured by streptavidin to form a lighter band at the T-line. It is important to note that the C-line was coated with goat anti-mouse IgG and could bind to colloidal gold-labeled FITC antibody regardless of the experiment. A test was judged invalid if there were no bands on the test strip. In this assay, results were defined as positive when a band at the T-line was not visible and a band at the C-line was visible, with negative results having visible bands at the T-line and C-line.

The RPA-CRISPR/Cas12a-LFA system for the *B1* gene was carried out using a commercial kit (EZassay, cat. No. D-L-CAS12-2S) according to the manufacturer's instructions. In

brief, the reaction comprised 10 µL of reaction buffer with a 1 µL DNA sample (with 1 µL of DEPC water as a blank/negative group), 0.5 µL of forward and reverse primers (20 µM), 2 µL of starter (10×), and 1 µL DEPC water, which was incubated at 37°C for 20 min. Then, 2 µL of the RPA reaction system and cleavage buffer (10×), 0.6 µL of reporter (4 µM), 1 µL of Cas12a protein (1 µM), crRNA (Cas12a, 1 µM), and 15.4 µL of DEPC water were transferred to the CRISPR/Cas12a cleavage assay. Reactions were carried out at 37°C for 30 min. Finally, the CRISPR/Cas12a-LFA detection reaction was diluted 1:10 in detection buffer, after which the samples were applied to the LFA strips (EZassay, cat. No. CS-FMBO-48) and incubated at room temperature for 5 min.

## Optimizing the results of the RPA-CRISPR/Cas12a-LFA using a digital visualization instrument

The RPA-CRISPR/Cas12a-LFA might produce false-negative results for weakly positive samples. To solve this problem, a digital visualization instrument (Helmen, Suzhou, China) was used to measure and read the depth of the T-line on the strip. The LFA T-line values of 10-fold serial dilutions of pMD-19T-*B1* were measured. Finally, the data generated were used to construct a fitting curve to determine the T-line value of weak positive samples.

## Using RPA-CRISPR/Cas12a-LFA to detect simulated *T. gondii* samples using a digital visualization instrument

The whole genome of *T. gondii* (initial concentration = 1.6 ng/µL) was continuously diluted using DEPC water by 10 and 2 times to create simulated samples for the CRISPR/Cas12a-LFA test. The digital visualization instrument was used to read the results for the different concentrations of the genome to identify the most intense RPA-CRISPR/Cas12a-LFA T-line of the simulated samples, including weak positive simulated samples.

## Detection in a mouse model of *T. gondii* infection by RPA-CRISPR/Cas12a technique

One thousand tachyzoites were counted using a cell counting plate and injected into each mouse as the experimental group (*n* = 3). The control group (*n* = 3) was injected intraperitoneally with the same volume of saline. At 1, 3, and 5 days, three mice from each group were selected for eyeball blood collection, and the blood samples were placed in anticoagulant tubes. Then, DNA was extracted from 200 µL of whole blood using a TIANamp Genomic DNA Kit (Tiangen, Beijing, China, Cat. No. DP304-03). We detected the *B1* gene in blood samples from the infected and control mice using the RPA-CRISPR/Cas12a-LFA and qPCR techniques.

## Application of the RPA-CRISPR/Cas12a-LFA system to investigate the prevalence of *T. gondii* in stray dogs and cats

A total of 248 blood samples from stray dogs and cats collected from Deqing, Wenzhou, Jiaxing, Zhoushan, Lishui, and Yiwu cities of Zhejiang province were used to detect *T. gondii* using the RPA-CRISPR/Cas12a and RPA-CRISPR/Cas12a-LFA systems established in our laboratory.

## ACKNOWLEDGMENTS

We would like to thank Zhejiang University for providing the DNA of *G. lamblia, C. parvum, E. bieneusi, and B. hominis*. We thank Dr. Jianhua Li at Jilin University for providing the DNA of *N. caninum*. We also thank the Zhejiang Small Animal Protection Association for their generous assistance during the collection of blood samples from stray cats and dogs.

This work was supported by the Key Projects Jointly Constructed by the Ministry and the Province of Zhejiang Medical and Health Science and Technology Project (grant

number WKJ-ZJ-2203), the Zhejiang Province "Pioneer," "Leading Goose" R&D Plan (grant number 2022C03109), the Central Leading Local Science and Technology Development Fund Project (grant number 2023ZY1019), the Health Commission of Zhejiang Province (grant number N2024KY923), and the Chinese Medicine Science and Technology Program of Zhejiang Province (grant number 2024ZL367).

H.S. performed the experiments, analyzed and interpreted the data, and drafted the manuscript. J. Fan, H.C., Y.G., Q.W., J. Fang, H.D., and X.Z. contributed to discussions and participated in some experiments. S.L. and B.Z. designed the experiments and reviewed the manuscript. All authors read and approved the final manuscript.

## AUTHOR AFFILIATIONS

[1]Laboratory of Pathogen Biology, School of Basic Medicine and Forensics, Hangzhou Medical College, Hangzhou, China
[2]Research Center of Novel Vaccine of Zhejiang Province, School of Basic Medicine and Forensics, Hangzhou Medical College, Hangzhou, China
[3]Key Laboratory of Bio-tech Vaccine of Zhejiang Province, School of Basic Medicine and Forensics, Hangzhou Medical College, Hangzhou, China

## AUTHOR ORCIDs

Hao Sun http://orcid.org/0009-0002-3333-3197
Xunhui Zhuo http://orcid.org/0000-0001-5805-0711
Bin Zheng http://orcid.org/0000-0003-3898-2946
Shaohong Lu http://orcid.org/0000-0001-9855-7154

## ETHICS APPROVAL

This study was approved by the Hangzhou Medical College Institutional Animal Care and Use Committee (Approval No.: 2023-038) and followed Chinese legislation regarding the use and care of research animals (GB/T35823-2018).

## ADDITIONAL FILES

The following material is available online.

Open Peer Review

**PEER REVIEW HISTORY (review-history.pdf).** An accounting of the reviewer comments and feedback.

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
