## [Reviewer comments · Microbiology Spectrum]

Microbiology Spectrum

RPA-CRISPR/Cas12a-LFA combined with a digital visualization instrument to detect *Toxoplasma gondii* in stray dogs and cats in Zhejiang province, China

Hao Sun, Jiyuan Fan, Hongkun Chu, Yafan Gao, Jiawen Fang, Qinli Wu, Haojie Ding, Xunhui Zhuo, Qingming Kong, Hangjun Lv, Bin Zheng, and Shaohong Lu

Corresponding Author(s): Hao Sun, Hangzhou Medical College

Review Timeline:

Submission Date:	November 30, 2023
Editorial Decision:	February 22, 2024
Revision Received:	April 4, 2024
Accepted:	April 14, 2024

Editor: Maristela Camargo

Reviewer(s): Disclosure of reviewer identity is with reference to reviewer comments included in decision letter(s). The following individuals involved in review of your submission have agreed to reveal their identity: Fang Wang (Reviewer #2)

Transaction Report:

DOI: <https://doi.org/10.1128/spectrum.03998-23>

Re: Spectrum03998-23 (RPA-CRISPR/Cas12a-LFA combined with a digital visualization instrument to detect *Toxoplasma gondii* in stray dogs and cats in Zhejiang province, China)

Dear Mr. Hao Sun:

Thank you for the privilege of reviewing your work. Below you will find my comments, instructions from the Spectrum editorial office, and the reviewer comments.

Revision Guidelines

Sincerely,
Maristela Camargo
Editor
Microbiology Spectrum

Reviewer #1 (Comments for the Author):

In this manuscript, Sun et al developed an RPA-CRISPR/Cas12a lateral flow band assay to diagnose *Toxoplasma* infection. The authors applied this assay to detect *Toxoplasma* infection in dogs and cats from Zhejiang province, China and found that the positive rate of the infection was 4%~8%. Overall, there are several major concerns that need to be addressed, and the manuscript is poorly written with many grammatical and typographical errors. Detailed comments are listed below:

1. The RPA-CRISPR/Cas12a assay based on the *Toxoplasma* B1 gene has been previously developed in other studies (Lei et al., ACS Synth. Biol., 2022; Wang et al., Parasites & Vectors, 2023). Given that the majority of the manuscript focuses on the method development, the current study seems simply to replicate the others' studies and apply the method to detect *Toxoplasma* infection in cats and dogs.
2. The optimization of this assay is mainly performed using a plasmid containing the B1 gene, which raises concerns about the reliability and sensitivity of this method when compared to other diagnostic methods.
3. In terms of the specificity of the method, the authors tested several parasite species that were phylogenetically distinct from *Toxoplasma*. Several coccidian parasites, such as *Neospora caninum* in dogs and *Hammondia hammondi* in cats, are commonly misdiagnosed as *Toxoplasma* in the clinical setting. The authors need to test if the assay can distinguish *Toxoplasma* from these two parasites.
4. The authors need to clarify the source of stray dog and cat samples. On lines 320~ 324, the author claims these are fecal samples. However, in Figure 1 and on line 428, the author states these are blood samples.
5. For all the experiments, what exactly are the negative controls?
6. A lot of references in this manuscript are not appropriate and accurate. Here is the list of these references: 1, 3, 7, 8, 12, 14, 15, 19, 25, 27, 30, 31, 35, 36, 37, and 39. Please change and correct these.
7. Please correct the typing errors in the manuscript, such as "high technical kills" on line 24, "eitht" on line 139
8. On line 63, please change "final host" to "definitive host".
9. On line 75, time-consuming is not a major flaw of microscopic detection. Generally, microscopic detection is not sensitive and accurate.
10. On line 102, the references do not have any information about *Cryptosporidium* and *Plasmodium*.

Reviewer #2 (Comments for the Author):

The study provided a rapid diagnosis method using RPA-CRISPR/Cas12a-LFA for early *T. gondii* infection detection, which has high accuracy and sensitivity without cross reaction with other tested parasites. Therefore, the has potential for investigation the prevalence of *T. gondii* infection in stray dogs and cats. But I strongly suggest the authors check spelling and grammar mistakes throughout the draft, and proofread the article by natural speakers.

1. Line 135: Correct the spelling mistake 'eitht' to 'eight'
2. Line 149: Correct the spelling mistake 'amont' to 'amount'
3. Line 164: The conclusion '(Figure 5B) showed that the lower limit of detection was 10 fg (31000 copies/ μ L)' cannot be tell from the figure 5B.
4. (Figure 5B) should be in Line 171
5. Figure 6C, explain why the fluorescence signal is lower than the qPCR result.
6. Line 211: Correct the spelling and grammar mistake: 'and the 14 postitive samples in RPA-CRISPR/Cas12a-LFA are correspond with RPA-CRISPR/Cas12a.'

Reviewers' comments:

Reviewer #1: Thanks for your professional review of our article. According to your suggestions, we have made extensive corrections to our previous draft, the details of which are listed below.

In this manuscript, Sun et al developed an RPA-CRISPR/Cas12a lateral flow band assay to diagnose *Toxoplasma* infection. The authors applied this assay to detect *Toxoplasma* infection in dogs and cats from Zhejiang province, China and found that the positive rate of the infection was 4%~8%. Overall, there are several major concerns that need to be addressed, and the manuscript is poorly written with many grammatical and typographical errors. Detailed comments are listed below:

1. The RPA-CRISPR/Cas12a assay based on the *Toxoplasma BI* gene has been previously developed in other studies (Lei et al., *ACS Synth. Biol.*, 2022; Wang et al., *Parasites & Vectors*, 2023). Given that the majority of the manuscript focuses on the method development, the current study seems simply to replicate the others' studies and apply the method to detect *Toxoplasma* infection in cats and dogs.

Re: Thank you for these comments. Compared with those two papers, the main innovations of our study are as follows:

Firstly, we established a visual RPA-CRISPR/Cas12a-LFA system combined with a digital visualization instrument, which minimized the problem of false-negative results for weakly positive samples and avoided misinterpretation of the results by the naked eye. In the studies by Lei et al. and Wang et al. study, they both used “elimination method” test strips. Li et al.^[1] report that the “elimination method” test strips are less sensitive and generate false-negative results.

Secondly, the developed RPA-CRISPR/Cas12a-LFA assay was used for a new epidemiological survey of the *Toxoplasma gondii* infection rate in stray cats and dogs in Zhejiang.

[1] Li H, Dong X, Wang Y, Yang L, Cai K, Zhang X, Kou Z, He L, Sun S, Li T, Nie Y, Li X, Sun Y. Sensitive and Easy-Read CRISPR Strip for COVID-19 Rapid Point-of-Care Testing. *CRISPR J.* 2021 Jun;4(3):392-399. doi: 10.1089/crispr.2020.0138. PMID: 34152219.

2. The optimization of this assay is mainly performed using a plasmid containing the *BI* gene, which raises concerns about the reliability and sensitivity of this method when compared to other diagnostic methods.

Re: Thank you for this comment. Plasmids have many advantages, such as convenience and stability, in the development of methods. It has been reported that it is feasible to verify the sensitivity of the detection method by constructing a *BI* gene plasmid. Before starting our study, we consulted previous studies. For example, Jiang et al.^[2] validated the sensitivity of the established PCR method by testing with the pMD-19T-*BI* plasmid, and they applied this method to detect *T. gondii* in meat.

Karakavuk et al.^[3] validated the sensitivity of the established LAMP method by testing the pCR 2.1-*RE* and pCR 2.1-*B1* plasmids, and they applied the developed LAMP method to detect *T. gondii* in cat feces.

Thank you for your suggestion. *Toxoplasma* positive DNA will be used in our future studies to obtain more realistic results.

[2] Jiang S, He Y, Zhang Y, et al. Assessment of self - produced PCR methods for the detection of *Toxoplasma gondii* DNA in meat [J]. *Journal of Food Safety*, 2016, 37(3).DOI 10.1111/jfs.12330

[3] Karakavuk M, Can H, Karakavuk T, Gül A, Alak SE, Gül C, Ün C, Gürüz AY, Döşkaya M, Döşkaya AD. Rapid detection of *Toxoplasma gondii* DNA in cat feces using colorimetric loop-mediated isothermal amplification (LAMP) assays targeting RE and B1 genes. *Comp Immunol Microbiol Infect Dis*. 2022 Feb;81:101745. doi: 10.1016/j.cimid.2022.101745. Epub 2022 Jan 6. PMID: 35030533.

3. In terms of the specificity of the method, the authors tested several parasite species that were phylogenetically distinct from *Toxoplasma*. Several coccidian parasites, such as *Neospora caninum* in dogs and *Hammondia hammondi* in cats, are commonly misdiagnosed as *Toxoplasma* in the clinical setting. The authors need to test if the assay can distinguish *Toxoplasma* from these two parasites.

Re: Thanks you for these comments. As shown in raw data figure, we also tested *Neospora caninum* using the RPA-CRISPR/Cas12a and RPA-CRISPR/Cas12a-LFA technology to validate the specificity of the method to detect *T. gondii*. We added this data to the resubmitted manuscript in lines 161, 166, 167, 322, 370 and in Figure 3F/5A. The sources of the relevant genomes have been added to the methods section. In China, there have been no studies reporting on *Hammondia hammondi*, and it is difficult to obtain it's DNA; therefore, we could not detect *Hammondia hammondi*.

4. The authors need to clarify the source of stray dog and cat samples. On lines 320~324, the author claims these are fecal samples. However, in Figure 1 and on line 428, the author states these are blood samples.

Re: We apologize for this error. It should be blood samples We have fixed the error in lines 325 and 327.

5. For all the experiments, what exactly are the negative controls?

Re: Thank you for this question. The negative controls contained water instead of the template DNA. In RPA-CRISPR/Cas12a detection, it is easy to cause aerosol contamination. In our study, using water as negative control could exclude aerosol contamination and avoid false positives. In the resubmitted manuscript, we describe the negative control in line 356.

6. A lot of references in this manuscript are not appropriate and accurate. Here is the list of these references: 1, 3, 7, 8, 12, 14, 15, 19, 25, 27, 30, 31, 35, 36, 37, and 39. Please change and correct these.

Re: We apologize for our carelessness. In our resubmitted manuscript, the erroneous references have been revised. Thanks for your correction.

7. Please correct the typing errors in the manuscript, such as "high technical kills" on line 24, "eitht" on line 139.

Re: All typing errors have been fixed.

8. On line 63, please change "final host" to "definitive host".

Re: Thank you for this comment. As suggested, "final host" has been corrected to "definitive host" in line 62.

9. On line 75, time-consuming is not a major flaw of microscopic detection. Generally, microscopic detection is not sensitive and accurate.

Re: Thank you for these comments. We have revised the sentence and its references to make it more accurate (line 75).

10. On line 102, the references do not have any information about *Cryptosporidium* and *Plasmodium*.

Re: We appreciate this valuable comment. We have added references dealing with *Cryptosporidium* and *Plasmodium* in the revised manuscript (line 104).

11. The manuscript is poorly written with many grammatical and typographical errors.

Re: We have improved the manuscript and invited the native English speaking scientists of Elixigen Company (Huntington Beach, California) to edit our manuscript. We hope the revised manuscript is acceptable.

Reviewer #2: Thank you your review of our manuscript. We have carefully considered all your comments and have revised our manuscript accordingly. All spelling and grammar errors have been corrected.

The study provided a rapid diagnosis method using RPA-CRISPR/Cas12a-LFA for early *T. gondii* infection detection, which has high accuracy and sensitivity without cross reaction with other tested parasites. Therefore, the has potential for investigation the prevalence of *T. gondii* infection in stray dogs and cats. But I strongly suggest the authors check spelling and grammar mistakes throughout the draft, and proofread the article by natural speakers.

1. Line 135: Correct the spelling mistake 'eitht' to 'eight'

Re: Thank you for this comment. We have corrected the 'eitht' to 'eight' in line 142.

2. Line 149: Correct the spelling mistake 'amont' to 'amount'

Re: Thank you for this comment. In our resubmitted manuscript, 'amont' has been revised to 'amount' in line 155.

3.Line 164: The conclusion '(Figure 5B) showed that the lower limit of detection was 10 fg (31000 copies/ μ L)' cannot be tell from the figure 5B.

Re: Thank you for this comment. In Figure 5B, we observed a low colored line in the T-line, and no definite band in the T-line for 100 fg. We apologize for our carelessness. As suggested by the reviewer, we have corrected 10 fg to 100 fg in line 171.

4.(Figure 5B) should be in Line 171

Re: Thank you for this excellent suggestion. We have corrected this error in line 172.

5.Figure 6C, explain why the fluorescence signal is lower than the qPCR result.

Re: Thank you for this comment. We used the SYBR Green method for qPCR detection. SYBR Green is a dsDNA-binding dye whose fluorescence increases up to 1000-fold upon intercalation with dsDNA. Moreover, as the amplification proceeds, the amount of DNA product increases, and hence the fluorescence of SYBR Green increases exponentially.

In the RPA-CRISPR/Cas12a assay, the fluorescent signal comes from the FAM probe in the reagent. As the positive gene amplifies, Cas12a begins to cleave the probe with FAM, allowing the FAM signal to be captured by the instrument. The FAM fluorescence does not increase exponentially. Hence, the fluorescence of FAM is lower than that in the qPCR assay.

In a previous study^[1] Figure 3C/3E show that the fluorescence of FAM is lower than that of the qPCR assay in the sensitivity detection of *T. gondii*.

[1] Lei R, Li L, Wu P, Fei X, Zhang Y, Wang J, Zhang D, Zhang Q, Yang N, Wang X. RPA/CRISPR/Cas12a-Based On-Site and Rapid Nucleic Acid Detection of *Toxoplasma gondii* in the Environment. *ACS Synth Biol.* 2022 May 20;11(5):1772-1781. doi: 10.1021/acssynbio.1c00620. Epub 2022 Apr 26. PMID: 35471824.

6. Line 211: Correct the spelling and grammar mistake: 'and the 14 positive samples in RPA-CRISPR/Cas12a-LFA are correspond with RPA-CRISPR/Cas12a.'

Re: We apologize for this error, which had been corrected in the revised manuscript in line 218.

7. I strongly suggest the authors check spelling and grammar mistakes throughout the draft, and proofread the article by natural speakers.

Re: We have improved the manuscript and invited the native English speaking scientists of Elixigen Company (Huntington Beach, California) to edit our manuscript. We hope the revised manuscript is acceptable.

Re: Spectrum03998-23R1 (RPA-CRISPR/Cas12a-LFA combined with a digital visualization instrument to detect *Toxoplasma gondii* in stray dogs and cats in Zhejiang province, China)

Dear Mr. Hao Sun:

Your manuscript has been accepted, and I am forwarding it to the ASM production staff for publication. Your paper will first be checked to make sure all elements meet the technical requirements. ASM staff will contact you if anything needs to be revised before copyediting and production can begin. Otherwise, you will be notified when your proofs are ready to be viewed.

Sincerely,
Maristela Camargo
Editor
Microbiology Spectrum